# Exploring Deep-Sea Biodiversity in the Porcupine Bank (NE Atlantic) through Fish Integrative Taxonomy

Rafael Bañón [1,2,*], Alejandro de Carlos [3], Carlos Farias [4], Nair Vilas-Arrondo [5,6] and Francisco Baldó [4]

1   Servizo de Planificación, Consellería do Mar, Xunta de Galicia, Rúa dos Irmandiños s/n,
    15701 Santiago de Compostela, Spain
2   Grupo de Estudos do Medio Mariño (GEMM), Edif. Club Naútico Bajo, 15960 Ribeira, Spain
3   Departamento de Bioquímica, Xenética e Inmunoloxía, Facultade de Bioloxía, Universidade de Vigo,
    Campus Universitario Lagoas-Marcosende s/n, 36310 Vigo, Spain; adcarlos@uvigo.es
4   Centro Oceanográfico de Cádiz, Instituto Español de Oceanografía (IEO, CSIC), Puerto Pesquero, Muelle de
    Levante s/n, 11006 Cádiz, Spain; carlos.farias@ieo.es (C.F.); francisco.baldo@ieo.es (F.B.)
5   AQUACOV, Centro Oceanográfico de Vigo, Instituto Español de Oceanografía (IEO, CSIC), Subida a Radio
    Faro 50-52, 36390 Vigo, Spain; nair_vilasarrondo@hotmail.com
6   Faculty of Biology, University of Vigo, 36310 Vigo, Spain
*   Correspondence: anoplogaster@yahoo.es

**Abstract:** This study combined morphological and molecular approaches to the species assignment of several rare or poorly known deep-water fishes caught between 549 and 1371 m depth during a Spanish bottom trawl survey in the Porcupine Bank, west of Ireland. The following fish species were identified: *Nessorhamphus ingolfianus* (Schmidt, 1912), *Borostomias antarcticus* (Lönnberg 1905), *Scopelosaurus lepidus* (Krefft and Maul 1955), *Bathypterois dubius* Vaillant, 1888, *Evermannella balbo* (Risso, 1820), *Antimora rostrata* (Günther, 1878), *Melanonus zugmayeri* Norman, 1930, *Lyconus brachycolus* Holt and Byrne, 1906; *Paraliparis hystrix* Merrett, 1983, *Neocyttus helgae* (Holt and Byrne, 1908); *Platyberyx opalescens* Zugmayer, 1911; *Howella atlantica* Post and Quéro, 1991, *Lycodes terraenovae* Collett, 1896 and *Pseudoscopelus altipinnis* Parr, 1933. The presence of *L. brachycolus*, *P. opalescens* and *P. altipinnis* is reported for the first time in the Bank. The DNA barcoding results were largely consistent with morphological identification in 10 species but four did not fit the current taxonomy, indicating cases of potential cryptic speciation, misidentification, synonymy or recent diversification. Among them, the results strongly suggest that *P. garmani* and *P. hystrix* are conspecific, making *P. hystrix* a junior synonym of *P. garmani*.

**Keywords:** NE Atlantic; distribution; taxonomic identification; morphology; DNA barcoding; synonymy

## 1. Introduction

The deep ocean makes up 95% of the volume of the seas and is the largest and least explored biome on the planet [1]. Organisms in this environment have evolved to adapt to high pressures, almost perpetual darkness and low food availability that characterizes much of the deep sea. The term deep-sea fishes was often used to refer to fish that live in darkness below the sunlit surface waters, at approximately 200 m water depth, i.e., below the epipelagic or photic zone of the sea or the continental shelf break. However, there is not rigid definition, and there is a lack of consensual criteria for establishing the initial boundary of deep-sea habitat. For example, Glover et al. [2] defined a deep-sea species as one that is more than 500 m deep, since at this distance from the surface the seasonal variation of physical parameters and the influence of sunlight are minimal.

As deep-sea fishes have been poorly sampled globally, overall knowledge of fish distributions and drivers of community composition and diversity remain incomplete [3]. The capture of rare or poorly known deep-sea fishes is of special interest as it adds to knowledge of the basic taxonomy, biology and distribution [4]. Such records allow a better

understanding of species distribution ranges and morphological variations, as well as the mechanisms involved in connectivity between areas [5,6].

'Integrative taxonomy' emerges as an important tool for species delimitation and thus for a better understanding of the composition of deep-sea ichthyofauna. It is defined as the science that aims to delimit the units of life's diversity from multiple and complementary perspectives [7]. DNA barcoding is a tool that has been successfully integrated and enhanced traditional morphological analysis in the systematic studies of fishes [8,9]. This integrative study can highlight identification mistakes and incongruities between molecular and morphological results, helping to reveal cryptic species, the identification of immature specimens, and clarification of problems of synonymy [10]. Gaps and inconsistencies in reference DNA databases can make accurate identification of fishes to species level difficult, suggesting the need of reinforcing DNA barcoding reference datasets [11]. Without reference sequences from voucher specimens identified by qualified taxonomists, there is no reliable library for comparing newly generated query sequences [12].

Conditions in deep-sea ecosystems are more uniform and constant than those in shallower waters, and this is the main reason given for the wider observed global distribution of deep-sea fishes. Under similar selective pressures, deep-sea fishes have convergently evolved adopting similar morphology, often developing analogous structures as adaptations to similar environments [13]. This phenomenon makes it difficult to correct identification of these fishes, which is an obstacle to a complete understanding of the true biodiversity. Molecular taxonomy has revealed cases of intraspecific divergence compatible with different species, exposing cases of possible cryptic species or the resurrection of synonyms [14].

The Porcupine Bank is located in the north-eastern Atlantic, 200 km off the west coast of Ireland, forming a seamount-like structure, with its related anticyclonic structures. The fish fauna of this bank and the adjacent areas are well reported in the ichthyological literature [15,16], but the occurrence of unreported fishes is not unusual [8,17], showing that the knowledge of this environment is far from being complete.

The aim of this manuscript is, firstly, to document the presence of poorly known deep-water fishes in the Porcupine Bank and secondly, to confirm the taxonomy of these species by means of an integrative taxonomy approach, combining both morphological examination and the molecular DNA barcoding method. These objectives are within the recommendations to inform the development of the Decadal Ocean Actions focused on the deep sea and help to resolve the question of what is the diversity of life in this environment? [18]. In particular, in the objective 1 "Capacity development", it is stated that all actions should commit to sharing specimens, including whole animals, tissue, barcoding and environmental DNA samples, and investing in the deposition of specimens in established and regionally relevant institutions that have recognized charters to support the permanent storage and care of archived specimens and the recommendation of open access publication of research and data whenever possible [19].

## 2. Materials and Methods

### 2.1. Study Site and Sampling

A Spanish bottom trawl research survey has been carried out annually since 2001 in the Porcupine Bank (ICES Divisions 7c and 7k), on board the R/V Vizconde de Eza, to study the distribution, relative abundance and biological parameters of commercial fish. The survey covers an area that extends from longitude 12° W to 15° W and from latitude 51° N to 54° N, following the standard methodology for the IBTS North Eastern Atlantic Surveys. In September–October 2020, during the 2020 Spanish Bottom Trawl Survey on the Porcupine Bank (SP-PORC-Q3) a total of 91 bottom trawls of 20 min of duration were made between 190 and 1400 m depth using a Baca-GAV 39/52 with a cod-end mesh size of 20 mm (Figure 1). Twenty-five specimens caught between 549 and 1371 m depth were selected for further study. After removing tissue samples for molecular analysis, the specimens, were

stored at −28 °C and deposited in the fish collection of the Museo Luis Iglesias de Ciencias Naturais in Santiago de Compostela (MHNUSC).

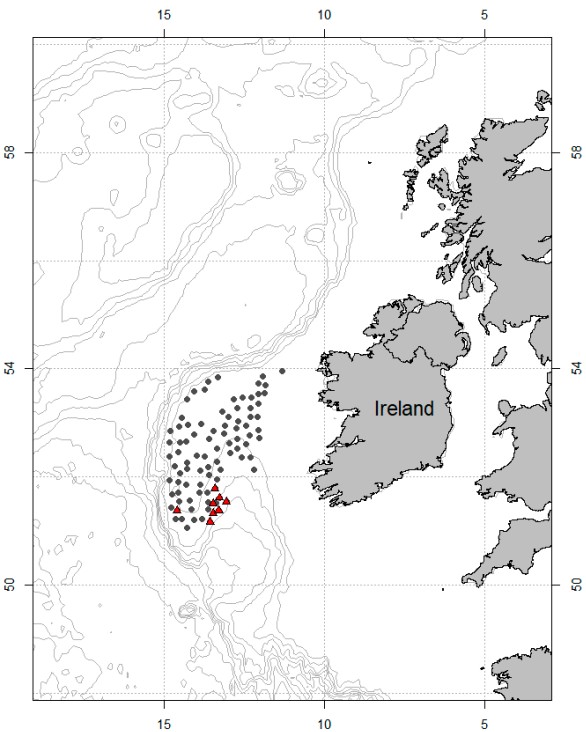

**Figure 1.** Map showing the bottom trawls (grey dots) conducted during the science cruise on the Porcupine Bank. The red triangle marks represent the locations where the analyzed specimens were collected.

### 2.2. Morphological Analysis

First, a preliminary onboard identification was carried out on fresh specimens. After the scientific survey, specimens were defrosted and definitively identified to species level following ichthyological guides and keys [20,21]. The main morphometric and meristic characters were recorded according to the literature as follows: Total length (TL), Standard length (SL), Head length (HL); Upper jaw length (JL); Pre-orbital length (PO); Eye diameter (ED); Post orbital Length (POL); Inter-orbital length (IO); Predorsal length (PD); Pre-first dorsal length (PD1); Pre-second dorsal length (PD2); Prepectoral length (PP); Pre-anal length (PA); Dorsal fin base length (DB); First dorsal fin base length (DB1); Second dorsal fin base length (DB2); Anal fin base length (AB); Pre-pectoral length (PP); Pre-pelvic length (PV); Pectoral fin length (PL); Pelvic fin length (VL); Maximum body depth (BH); Caudal peduncle depth (CP); Number of rays in dorsal fin (D); Number of rays in first dorsal fin (D1); Number of rays in second dorsal fin (D2); Number of rays in pectoral fin (P); Number of rays in anal fin (A); Number of rays in caudal fin (C); Branchiostegal rays (BR); Gill-rakers (GR); Scales in the lateral line (SLL). With the exception of TL and SL, measurements are distances perpendicular to the length of the fish measured with a digital calliper to the nearest mm. All measurements are expressed as the percentage of standard length (%SL). Descriptive data are reported individually for species represented by one or two specimens and ranges are reported when there are three or more.

### 2.3. DNA Extraction, PCR Amplification and Sequencing

Molecular procedures were carried out in the Biochemistry laboratories of the University of Vigo (Spain). DNA extraction and purification were carried out from muscle tissue of each specimen, using the E.Z.N.A. Tissue DNA Kit from Omega Bio-Tek, following the manufacturer's instructions. About 30 mg of muscle tissue was used and total

DNA was recovered in 200 μL of elution buffer. The primer sets used in the sequence amplification procedures were the COI-1 or COI-3 cocktails for the COI-5P barcoding marker [22] and, in the case of *Lycodes terraenovae*, 12SA-12SB for 12S rRNA and GLU-CB2 for Cyt b [23]. All information regarding these specimens as well as their DNA barcodes, images, places of capture and other complementary data are available in the project "Fishes of the Porcupine Bank" (code PORCU) in the Barcoding of Life Database (BOLD, http://www.boldsystems.org/ (accessed on 6 September 2021)).

Each PCR reaction was carried out in 20 μL final volume including Phire Green Hot Start II PCR Master Mix (Thermo Scientific, Göteborg, Sweden) for COI-5P amplifications, or Green-Taq DNA polymerase Master Mix (Canvax), the corresponding primers, nuclease-free water and 2 μL of template DNA. The time–temperature profiles were for COI: 30 s at 98 °C; 35 cycles of 5 s at 98 °C, 10 s at 52 °C and 15 s at 72 °C; 1 min at 72 °C. For Cyt b and 12 S: 2 min at 94 °C; 30 cycles of 1 min at 94 °C, 1 min at 52 °C and 1 min at 72 °C; 1 cycle of 30 s at 94 °C, 30 s at 52 °C and 5 min at 72 °C. The products of the PCR reactions were visualized by electrophoresis on 2% E-Gel with SYBR Safe on an E-Gel Power Snap apparatus (Invitrogen), and sent to the Genomics Service of the Centre for Scientific and Technological Support to Research (CACTI) of the University of Vigo, for subsequent purification and sequencing by the Sanger method [24]. The sequencing reactions were carried out in both senses to subsequently generate a consensus sequence, using the same primers as in the amplifications, except for the amplicons obtained using the COI-3 cocktail, where the primers M13F (-21) and M13R (-27) [25] were used.

### 2.4. Molecular Analysis and Assignment of Specimens

For each of the 14 taxa, the COI-5P sequences of the specimens assigned after morphological analysis to the species level were used as queries in the nucleotide BLAST (BLASTn) tool (https://blast.ncbi.nlm.nih.gov/Blast.cgi (accessed on 6 September 2021)). In those cases, where the percentage of identity was low (less than 98%) or the resulting species name did not match, the sequences were manually aligned with others belonging to the same genus that were publicly available in the BOLD database. Sequences were collapsed into unique haplotypes employing FaBox [26]. Molecular taxonomy cladograms were produced by the Neighbor-Joining (NJ) method [27] using MEGA X [28]. Observed differences between sequences were expressed proportionally as p-distances [29]. Despite their purely taxonomic use, a bootstrap resampling process [30] was included in the elaboration of the trees, with 2000 iterations. A total of 652 nucleotide positions were considered in the final alignments, with a gap/missing data treatment of pairwise deletion.

Species recognition using barcodes relies on different species having different unique sequences or different assemblages of closely related sequences so that the intraspecific variation or genetic distance is thus generally much less than interspecific variation, enabling species identification [31].

### 3. Results

### 3.1. Morphological Traits

A total of 25 specimens in 14 families were identified, comprising 14 species (Figure 2).

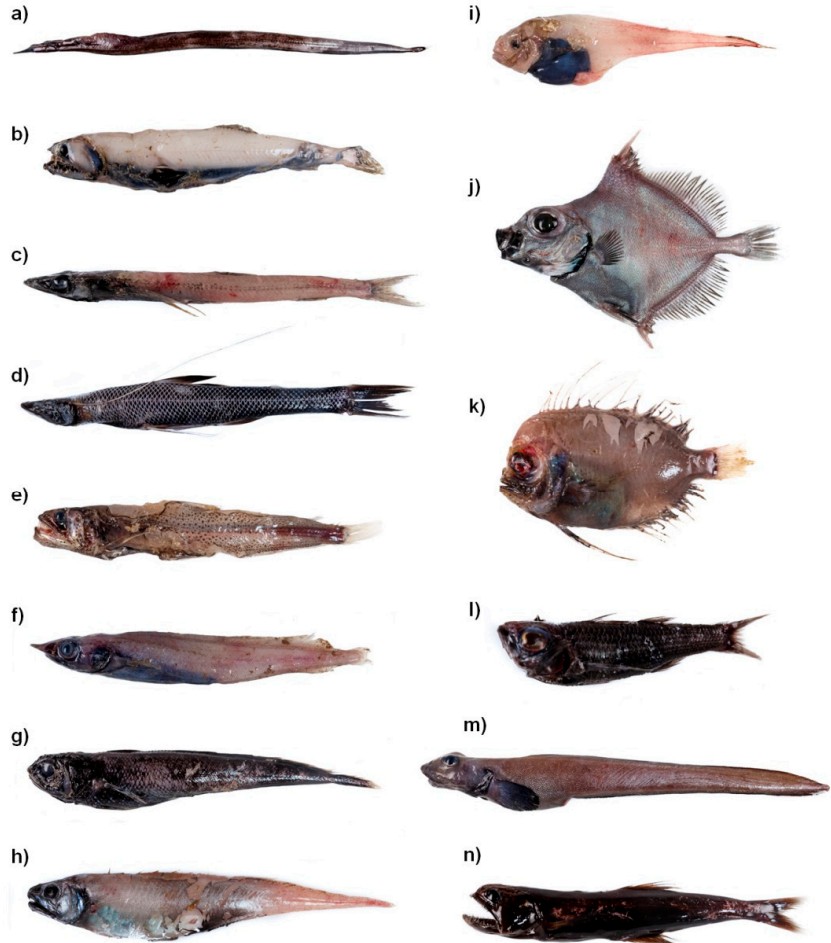

**Figure 2.** Deep-water fishes caught on the Porcupine Bank: (**a**) *N. ingolfianus* 462 mm TL; (**b**) *B. antarcticus* 266 mm TL; (**c**) *S. lepidus* 148 mm TL; (**d**) *B. dubius* 174 mm TL; (**e**) *E. balbo* 113 mm TL; (**f**) *A. rostrata* 107 mm TL; (**g**) *M. zugmayeri* 222 mm TL; (**h**) *L. brachycolus* 281 mm SL; (**i**) *P. hystrix* 106 mm TL; (**j**) *N. helgae* 142 mm TL; (**k**) *P. opalescens* 97 mm TL; (**l**) *H. atlantica* 100 mm TL; (**m**) *L. terraenovae* 365 mm TL; (**n**) *P. altipinnis* 136 mm TL (Photos: Francisco Baldó).

### 3.1.1. *Nessorhamphus ingolfianus* (Schmidt, 1912) (Anguilliformes: Derichthyidae)

Material examined: MHNUSC 25179-1, 111 g, 538 mm TL, 525 mm SL, 13 October 2020, 51.544° N, −13.0532° W, 1371 m depth; MHNUSC 25179-2 (Figure 2a), 59 g, 462 mm TL, 450 mm SL, 13 October 2020, 51.544° N, −13.0532° W, 1371 m depth. Morphology: HL: 11.3 and 11.1; PO: 4.9 and 5.2; POL: 5.3 and 4.9; ED: 1.2 and 1.1; IO: 0.5 and 0.6; PD: 19.4 and 20.7; DB: 38.7 and 40.9; PA: 58.1 and 57.3; AB: 38.7 and 40.9; PP: 13 and 11.8; PL: 3 and 3.3; BH: 5.2 and 4.9; P: 12 and 12. Distribution: Circumglobal in tropical and warm temperate seas. In the eastern Atlantic it occurs from Iceland and Irminger Sea, to Morocco and off the Cape, South Africa [20,32].

### 3.1.2. *Borostomias antarcticus* (Lönnberg, 1905) (Stomiiformes: Stomiidae)

Material examined: MHNUSC 25180-1, 23 g, 178 mm TL, 164 mm SL, 13 October 2020, 51.6198° N, −13.2722° W, 986 m depth; MHNUSC 25180-2, 78 g, 250 mm TL, 235 mm SL, 13 October 2020, 51.6198° N, −13.2722° W, 986 m depth; MHNUSC 25180-3 (Figure 2b), 81 g, 266 mm TL, 245 mm SL, 13 October 2020, 51.544° N, −13.0532° W, 1371 m depth; MHNUSC 25180-4, 102 g, 252 mm TL, 239 mm SL, 13 October 2020, 51.544° N, −13.0532° W, 1371 m depth. Morphology: HL: 19.2–20.4; PO: 5.3–5.5; POL: 11–11.9; ED: 2.6–3.3; IO: 4.9–5.4; PD: 54.7–56.9; DB: 9–10; PA: 77.1–79.6; AB:12.8–13.9; PP: 17–20.9; PV: 52.7–57.4; PL: 10.6–11.9; BH: 10.2–16.3; D: 12; A: 16–17; P: 8–9; V: 7. Distribution: Widely distributed in all oceans;

in the northeast Atlantic it is found from Greenland and Iceland southwards to about 42° N [20].

### 3.1.3. *Scopelosaurus lepidus* (Krefft and Maul, 1955) (Aulopiformes: Notosudidae)

Material examined: MHNUSC 25181 (Figure 2c), 8 g, 148 mm TL, 129 mm SL, 13 October 2020, 51.544° N, −13.0532° W, 1371 m depth. Morphology: HL: 26.4; PO: 8.5; JL: 13.2; POL: 13.2; ED: 5.4; IO: 3.9; PD: 54.3; DB: 7.0; PA: 79.1; AB: 10.9; PP: 27.9; PV: 47.3; PL: 27.1; VL: 10.9; BH: 7.8; D: 11; A: 18; P: 14; V: 9; Gr: 2+17. Distribution: North Atlantic, from Greenland, Iceland and Faroe-Iceland Ridge to Florida, Grand Meteor Seamount and Mauritanian Upwelling Region to about 15° N. Scattered records in central equatorial, southward to about 10° S [20].

### 3.1.4. *Bathypterois dubius* Vaillant, 1888 (Aulopiformes: Ipnopidae)

Material examined: MHNUSC 25182 (Figure 2d), 17 g, 174 mm TL, 140 mm SL, 13 October 2020, 51.544° N, −13.0532° W, 1371 m depth. Morphology: HL: 19.6; PO: 6.6; POL: 10; ED: 3.9; IO: 2.4; PD: 22.1; DB: 47.9; PA: 62.9; AB: 15.7; PP: 18.6; PV: 39.3; PL: 75; VL: 24.9; BH: 13.6; D: 16; A: 10; P: 12; V: 8; Gr: 13+1+23; SLL: 56. Distribution: Eastern Atlantic Ocean from the British Isles to Sierra Leone, Azores and the Mediterranean Sea; one record from the western North Atlantic Ocean [32].

### 3.1.5. *Evermannella balbo* (Risso, 1820) (Aulopiformes: Evermannellidae)

Material examined: MHNUSC 25183 (Figure 2e), 6 g, 113 mm TL, 100 mm SL, 14 October 2020, 51.1731° N, −13.5604° W, 1037 m depth. Morphology: HL: 17.1; JL: 14; PO: 5.7; POL: 7.3; ED: 4.1; IO: 1; PD: 45; DB: 9; PA: 65; AB: 25.5; PP: 23; PV: 46; PL: 14; VL: 8.5; BH: 17; CP: 5.1; D: 14; A: 35; P: 11; V: 8. Distribution: In the North Atlantic from southern Iceland to about 30° N and along the East Atlantic to equatorial waters. Mediterranean Sea. Rare in the Caribbean Sea. Associated with the Subtropical Convergence in the South Atlantic, Indian and Pacific Oceans [20].

### 3.1.6. *Antimora rostrata* (Günther, 1878) (Gadiformes: Moridae)

Material examined: MHNUSC 25184 (Figure 2f), 6 g, 107 mm TL, 104 mm SL, 13 October 2020, 51.544° N, −13.0532° W, 1371 m depth. Morphology: HL: 26.4; PO: 7.7; POL: 13.5; ED: 5.3; IO: 3.3; PD1: 29.8; PD2: 32.7; DB1: 2.9; DB2: 61.5; PA: 57.7; AB: 34.6; PP: 27.9; PV: 22.1; PL: 13.5; VL: 16.3; BH: 15.4; D1: 4; D2: 52; A: 36; P: 20; V: 6; GR: 4+1+12. Distribution: Circumglobal, except in the North Pacific; in the eastern Atlantic, from Iceland to South Africa [32].

### 3.1.7. *Melanonus zugmayeri* Norman, 1930 (Gadiformes: Melanonidae)

Material examined: MHNUSC 25185-1, 54 g, 220 mm TL, 209 mm SL, 14 October 2020, 51.5101° N, −13.4684° W, 835 m depth; MHNUSC 25185-2 (Figure 2g), 59 g, 222 mm TL, 214 mm SL, 14 October 2020, 51.1731° N, −13.5604° W, 1037 m depth; MHNUSC 25185-3, 17 g, 154 mm TL, 148 mm SL, 13 October 2020, 51.1731° N, −13.5604° W, 1037 m depth. Morphology: HL: 18.7–19.1; PO: 4.7–5.3; POL: 9.5–10; ED: 3.7–4.7; IO: 7.2–7.5; PD: 22.9–23.6; DB: 60.8–64; PA: 46.7–54.7; AB: 33.1–40.2; PP: 20.9–21.1; PV: 17.6–20.1; PL: 12.8–16.4; VL: 13.9–14.5; BH: 14.2–16.4; CP: 1.4–1.9; D: 82; A: 56–61; P: 15–17; V: 7; BR: 6; GR: 3+6–7. Additional characters: 12–15 vomer teeth. Distribution: Circumglobal in tropical and subtropical seas [32]; in the eastern Atlantic from about 60° N to South Africa [20].

### 3.1.8. *Lyconus brachycolus* Holt and Byrne, 1906 (Gadiformes: Lyconidae)

Material examined: MHNUSC 25186-1 (Figure 2h) (tip broken), 91 g, 281 mm SL, 1 October 2020, 51.7917° N, −13.4086° W, 741 m depth; MHNUSC 25186-2, 34 g, 180 mm TL (tip amputated), 14 October 2020, 51.3335° N, −13.4631° W, 1025 m depth. Morphology: HL: 17.1 and 20.6; PO: 5 and 6.1; POL: 7.8 and 10; ED: 4.3 and 4.4; IO: 5.7 and 5.6; PD1: 23.1 and 23.9; DB1: 4.3 and 5; PA: 47.3 and 50.6; PP: 17.8 and 21.1; PV: 18.5 and 22.2; PL: 11 and

24.4; VL: 4.6 and 6.7; BH: 13.2 and 15.6; D1: 10 and 9; P: 14-13; V: 9; BR: 7; GR: 3+10 and 3+11. Distribution: North Atlantic from about 60° N of Scotland to about 10° N, also off Namibia [20].

### 3.1.9. *Paraliparis hystrix* Merrett, 1983 (Scorpaeniformes: Liparidae)

Material examined: MHNUSC 25187-1, 4.8 g, 117 mm TL, 106 mm SL, 14 October 2020, 51.510° N, −13.468° W, 835 m depth; MHNUSC 25187-2 (Figure 2i), 5.1 g, 106 mm TL, 99 mm SL, 14 October 2020, 51.173° N, −13.560° W, 1037 m depth; MHNUSC 25187-3, 6.8 g, 110 mm TL, 100 mm SL, 14 October 2020, 51.173° N, −13.560° W, 1037 m depth; MHNUSC 25187-4, 5.4 g, 99 mm TL, 88 mm SL, 14 October 2020, 51.173° N, −13.560° W, 1037 m depth. Morphology: HL: 20–21.8; JL: 9.9–11.3; PO: 6.4–7.7; POL: 9.4–10.7; ED: 3.4–4.2; IO: 4.7–7; PD: 24.3–27; DB: 68.4–75.2; PA: 33.8–40.5; AB: 59.4–65.4; PP: 14.3–20; PL (upper lobe): 11.3–13.4; BH: 19–24.3; D: 53–58; A: 46–49; P (upper lobe+notch+lower lobe): 11–14+2–4+2–4; C: 7–8; BR: 6; GR: 0+6–7. Distribution: Eastern North Atlantic, to southwest of Ireland [33].

### 3.1.10. *Neocyttus helgae* (Holt and Byrne, 1908) (Zeiformes: Oreosomatidae)

Material examined: MHNUSC 25188 (Figure 2j), 45 g, 142 mm TL, 120 mm SL, 15 October 2020, 51.3893° N, −13.2932° W, 1162 m depth. Morphology: HL: 34.2; JL: 8.3; PO: 8.3; POL: 10; ED: 15.8; IO: 10; PD: 43.3; DB: 41.7; PA: 50.8; AB: 34.2; PP: 35; PV: 37.5; PL: 15.8; VL: 15; BH: 50.8; CP: 5.8; D: VII+34; A: III+31; P: 18; V: I+6; C: 2+13+2; BR: 6; GR: 4+16. Additional characters: 2nd dorsal spine length: 18.3% SL; 1st anal spine length 16.7% SL; pelvic spine length: 15% SL. Distribution: Northeastern Atlantic, from Iceland to Madeira and North-western Atlantic [32].

### 3.1.11. *Platyberyx opalescens* Zugmayer, 1911 (Perciformes: Caristiidae)

Material examined: MHNUSC 25189 (Figure 2k), 22 g, 97 mm TL, 80 mm SL, 14 October 2020, 51.5101° N, −13.4684° W, 835 m depth. Morphology: HL: 37.5; JL: 16.3; PO: 3.8; POL: 15; ED: 18.8; IO: 7.5; PD: 21.3; DB: 71.3; PA: 53.8; AB: 36.3; PP: 37.5; PV: 25; PL: 31.3; VL: 45; BH: 55; CP: 12.5; D: 28; A: 19; P: 20; V: 6; C: 2+20+2; GR: 8+14. Additional characters: suborbital space 4.4% SL; suborbital series not expanded to cover upper jaw; upper jaw relatively long, extending beyond mid-orbit; 9 palatine and 23 vomerine teeth. Distribution: Eastern Atlantic, from Greenland and Iceland to Namibia and Central Atlantic, in Azores and Mid-Atlantic Ridge [20,34].

### 3.1.12. *Howella atlantica* Post and Quéro, 1991 (Perciformes: Howellidae)

Material examined: MHNUSC 25190 (Figure 2l), 12 g, 100 mm TL, 84 mm SL, 19 September 2020, 51.3801° N, −14.5886° W, 549 m depth. Morphology: HL: 34.5; PO: 8.3; JL: 13.1; POL: 14.3; ED: 11.9; IO: 9.2; PD1: 36.9; PD2: 60.7; DB1: 11.9; DB2: 9.5; PA: 64.3; AB: 7.1; PP: 35.7; PV: 34.3; PL: 34.5; VL: 17.3; BH: 27.4; D1: 8; D2: I+9; A: III+7; P: 15; V: I+5; BR: 7; GR: 7+20; SLL: 3+8+27. Additional characters: Three rows of scales from lateral line to second dorsal-fin origin; first dorsal-fin spine minute, its length (1 mm) was the 10% the length of second spine (10 mm). Distribution: Western and Eastern Atlantic Ocean, from 64° N to 20 °S [35].

### 3.1.13. *Lycodes terraenovae* Collett, 1896 (Perciformes: Zoarcidae)

Material examined: MHNUSC 25191 (Figure 2m), 167 g, 365 mm TL, 360 mm SL, 15 October 2020, 51.3893° N, −13.2932° W, 1162 m depth. Morphology: HL: 18.6; PO: 5.3; POL: 9.4; ED: 3.9; IO: 3.3; PD: 25.3; DB: 74.7; PA: 38.1; AB: 61.9; PP: 18.9; PV: 15.8; PL: 11.7; VL: 2.2; BH (to anus level): 11.8; P: 22; V: 1; BR: 5; GR: 0+10. Additional characters: Double lateral line; submental crests not united at symphysis; pectoral fin rounded; scales present in the predorsal area and the abdominal part; 9 palatine teeth; 9 vomer teeth; 26 premaxillary teeth; spinules on gill-rakers of first gill arch; coloration uniformly brownish; peritoneum dark. Distribution: Both sides of the Atlantic Ocean; in eastern Atlantic from Rockall to South Africa [36]. Remarks: The stomach content reveals the presence of 7 whole discs

and a large number of arm-ossicles from brittle-stars belonging to the familiy Ophiactidae. There are also sponge-spicules from the hexactinellid *Pheronema carpenteri* (Thomson, 1869), which might be considered as accidental prey, because of this brittle-stars seem to live closely associated to this deep-sea sponge, well known as an important habitat-builder species.

### 3.1.14. *Pseudoscopelus altipinnis* Collett, 1896 (Perciformes: Chiasmodontidae)

Material examined: MHNUSC 25192-1 (Figure 2n), 15 g, 136 mm TL, 115 mm SL, 14 October 2020, 51.3335° N, −13.4631° W, 1025 m depth. MHNUSC 25192-2, 14 g, 126 mm TL, 106 mm SL, 1 October 2020, 51.7917° N, −13.4086° W, 741 m depth. Morphology: HL: 26.8 and 27.5; PO: 7.0 and 6.7; POL: 16.0 and 16.3; ED: 3.8 and 4.9; IO: 7.8 and 9.1; PD1: 33.0 and 33.0; PD2: 49.6 and 50.0; DB1: 9.6 and 13.2; DB2: 37.4 and 38.7; PA: 48.7 and 49.1; AB: 37.4 and 38.7; PP: 26.1 and 28.3; PV: 27.8 and 31.1; PL: 20.0 and 20.3; VL: 12.2 and 11.7; BH: 17.4 and 20.1; CP: 5.7 and 5.8; D1: VII; D2: I+25; A: II+24; P: 12; V: I+5; BR: 7. Distribution: Widespread in the Atlantic and Pacific Oceans [37].

### 3.2. NJ Trees and Genetic Distances

COI barcodes were obtained from all 25 specimens (Table S1). The four sequences of *P. histrix* and the one from *L. terraenovae* and *P. opalescens* are new contributions to the BOLD project.

Table 1 shows the result of the molecular identifications of the specimens using COI-5P sequences, compared to those made by morphology. In 10 of the 14 taxa involved, the identification is coincident. Where traditional taxonomy identifies *H. atlantica*, *L. terraenovae* and *P. opalescens*, BLASTn analysis returns *Howella broidei*, *Lycodes adolfi* and *Paracaristius maderensis*, respectively. In the case of the four specimens of *P. hystrix*, the percentage of identity with BLASTn is too low to be reliable, being in the 94–95% range, returning *Paraliparis* sp.

**Table 1.** Top total score BLASTn identification results of COI barcodes.

| Taxonomic ID | GenBank Acc. No. | BLASTn ID | Query Cover (%) | E-Value | Identity (%) | Accession |
|---|---|---|---|---|---|---|
| *Antimora rostrata* | MW907994 | *A. rostrata* | 100 | 0.0 | 100 | JF265134 |
| *Bathypterois dubius* | MW907995 | *B. dubius* | 99 | 0.0 | 99.85 | KC015247 |
| *Borostomias antarcticus* | MW907998 | *B. antarcticus* | 100 | 0.0 | 99.85 | KF929668 |
| *Borostomias antarcticus* | MW907996 | *B. antarcticus* | 100 | 0.0 | 99.85 | KF929668 |
| *Borostomias antarcticus* | MW907999 | *B. antarcticus* | 100 | 0.0 | 99.69 | KF929668 |
| *Borostomias antarcticus* | MW907997 | *B. antarcticus* | 100 | 0.0 | 99.54 | KY033872 |
| *Evermannella balbo* | MW908000 | *E. balbo* | 99 | 0.0 | 99.85 | KY033618 |
| *Howella atlantica* | MW908001 | *H. brodiei* | 100 | 0.0 | 99.69 | EU148199 |
| *Lycodes terraenovae* | MW908002 | *L. adolfi* | 100 | 0.0 | 99.69 | GU804885 |
| *Lyconus brachycolus* | MW908003 | *L. brachycolus* | 99 | 0.0 | 98.92 | EU148230 |
| *Lyconus brachycolus* | MW908004 | *L. brachycolus* | 99 | 0.0 | 98.38 | EU148230 |
| *Melanonus zugmayeri* | MW908005 | *M. zugmayeri* | 100 | 0.0 | 99.39 | EU148249 |
| *Melanonus zugmayeri* | MW908007 | *M. zugmayeri* | 100 | 0.0 | 99.85 | EU148249 |
| *Melanonus zugmayeri* | MW908006 | *M. zugmayeri* | 100 | 0.0 | 99.69 | EU148249 |
| *Neocyttus helgae* | MW908008 | *N. helgae* | 100 | 0.0 | 100 | EU148264 |
| *Nessorhamphus ingolfianus* | MW908009 | *N. ingolfianus* | 100 | 0.0 | 100 | EU148266 |
| *Nessorhamphus ingolfianus* | MW908010 | *N. ingolfianus* | 100 | 0.0 | 100 | EU148266 |
| *Paraliparis hystrix* | MW908014 | *Paraliparis* sp. | 99 | 0.0 | 94.93 | KX676118 |
| *Paraliparis hystrix* | MW908011 | *Paraliparis* sp. | 99 | 0.0 | 95.08 | KX676118 |
| *Paraliparis hystrix* | MW908012 | *Paraliparis* sp. | 100 | 0.0 | 95.09 | KX676118 |
| *Paraliparis hystrix* | MW908013 | *Paraliparis* sp. | 100 | 0.0 | 95.09 | KX676118 |
| *Platyberyx opalescens* | MW908015 | *P. maderensis* | 96 | 0.0 | 99.84 | EU148108 |
| *Pseudoscopelus altipinnis* | MW908016 | *P. altipinnis* | 100 | 0.0 | 98.93 | MT323772 |
| *Pseudoscopelus altipinnis* | MW908017 | *P. altipinnis* | 100 | 0.0 | 98.77 | MT323772 |
| *Scopelosaurus lepidus* | MW908018 | *S. lepidus* | 100 | 0.0 | 99.39 | KF930385 |

The molecular identification process of the COI-5P sequences of *P. hystryx* is improved in relation to the BLAST search, following the Neighbor-Joining analysis (Figure S1A). In this case, the resulting haplotypes cluster robustly with sequences from *P. garmani* and others identified only at the genus level, exhibiting a maximum distance between them of 0.0107 and 0.0428 from their nearest neighbor.

The alignment of sequence MW908015 assigned to *P. opalescens* (PORCU021-21) was constructed using the closest available sequences found in the BOLD database for the family Caristiidae (Figure S1B). The sequence groups with two other *Paracaristius maderensis* (Maul, 1949) from the North Atlantic into a statistically well-supported clade with a mean distance between sequences of 0.001, and 0.0401 to its nearest neighbor, FMVIC386 assigned to *Caristius macropus* (Bellotti, 1903). Overall, the NJ tree shows several statistically well-supported clusters that include sequences from different species and genera mixed together.

The cladogram in Figure S1C shows that the sequence MW908001 from *L. terraenovae* (PORCU023-21) clusters into a statistically well-supported clade with the available COI barcodes of the species *Lycodes adolfi* Nielsen and Fosså, 1993 and *Lycodes pallidus* Collett 1879, with a maximum genetic distance of 0.0061. When other mitochondrial marker sequences, such as 12S and Cyt b, are obtained from the same specimen, they cluster with existing sequences from *L. terraenovae* and *L. adolfi* (Figure S2).

## 4. Discussion

Although current ichthyological research is largely oriented towards ecological and applied ichthyological aspects, many basic scientific questions remain unsolved for many fish species. The taxonomic knowledge of marine fish is much higher for coastal and/or commercial species compared to non-commercial or deep-sea species [38]. Many deep-water fishes are thought to have wide geographical distributions, but for some species, this conclusion rests primarily on morphological traits [39]. DNA barcoding is an important tool for fish species identification, but also provides a standardized measure of sequence divergence between distant areas. Integrative morphological and molecular analysis will allow also to reveal geographical divergences [8]. This methodology has been successfully used to detect cases of misidentification, synonymies and crypticism [8,9,38] and is being progressively implemented in fish biodiversity studies.

In relation to the specimens captured on the Porcupine Bank, it can be said that, in general terms, the morphology (biometric and meristic data) are in agreement with previous reports in the ichthyological literature [20,21,36,37].

The correlation between traditional taxonomic identification and DNA barcoding of the 25 specimens and their assignment to valid species was successful in 10 of the 14 taxa.

Of the four divergent cases, *H. atlantica* inBLASTn analysis returned *H. brodiei* as the best result, but that identification could have been based on *Howella brodiei atlantica* Post and Quéro, 1991 now considered a synonym of *H. atlantica* [20].

The three haplotypes of *Paraliparis hystrix* clustered with two *Paraliparis garmani* Burke, 1912 and two *Paraliparis* sp. with interspecific distances at intraspecific levels, which could indicate potential synonymy. *Paraliparis garmani* was described off New England, western Atlantic, by Burke [40] and then was included in the revision of the family [41], and *P. hystrix* was described decades after by Merret [42] in the west and south of Ireland, in the eastern Atlantic. Interestingly, Merrett [42] examining the holotype of *P. garmani* as comparative material, which leads us to conclude that he thought it was the closest species. Despite the high level of overlap found in morphological characters between the two species (Table 2), there is no comparative analysis in Merret's manuscript and this similarity has not been highlighted. No additional descriptions or identification keys to confirm the validity of the two species have been published. DNA barcoding not only serves to detect overlooked species of liparids, but also to rapidly generate important insights into the taxonomy and diversification of this group [43]. The results of this research strongly suggest based on morphology and molecular, *P. hystrix* is a junior synonym of *P. garmani*.

**Table 2.** Comparative morphological characters between *P. garmani* and *P. hystrix*.

| | *Paraliparis garmani* [40,41] | *Paraliparis hystrix* [42] |
|---|---|---|
| Contained in standard length | | |
| Head length | 4.5 | 4.2–5.7 |
| Body depth | 4.6 | 3.9–5.1 |
| Contained in head length | | |
| Eye diameter | 5.4 | 2.3–3.4 |
| Snout length | – | 2.2–3.4 |
| Gill slit length | 4.2 | 4.7 |
| Upper pectoral lobe length | 2 | 1.3–1.8 |
| Counts | | |
| Dorsal fin rays | 54 | 51–60 |
| Upper pectoral fin rays | 14 | 11–15 |
| Notch fin rays | 4 | 2–5 |
| Lower pectoral fin rays | 3–4 | 2–3 |
| Total pectoral fin rays | 21 | 16–21 |
| Caudal fin rays | 8 | 7–8 |
| Pyloric caeca | 6 | 4–8 |

Geographic distribution also supports this hypothesis (Figure 3). Chernova et al. [33] report the distribution of each species, eastern Atlantic for *P. hystrix* and western Atlantic for *P. garmani*. However, there are records of both species across both areas [44–46] indicating a single widespread species in the North Atlantic.

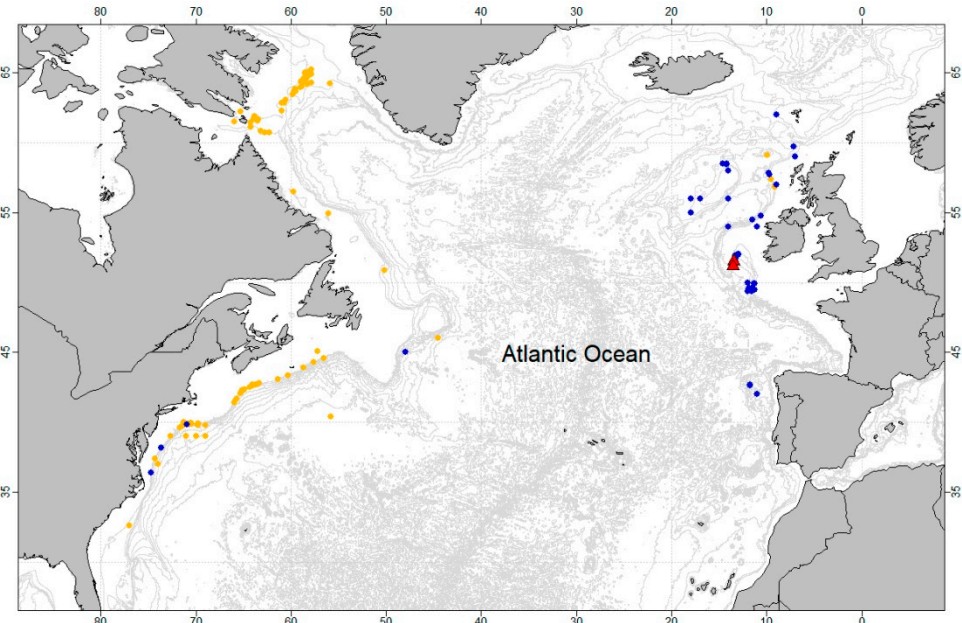

**Figure 3.** Distribution map showing the records of *P. hystrix* (blue dots) and *P. garmani* (yellow dots) reported by GBIF and OBIS, and the literature reviewed. The red triangle marks represent the new locations of *P. hystrix* reported in this research.

The sequence of *Platyberyx opalescens* clustered with two MAECO sequences of *P. maderensis* from the Mid-Atlantic Ridge. However, *P. opalescens* can be distinguished from *P. maderensis* by the suborbital series not expanded, space between orbit and mouth narrow; upper jaw relatively long, extending to posterior margin of orbit and the presences of palatine and vomerine teeth [34]. Additionally, the morphological description is in general agreement with that of Stevenson and Kenalay [47]. Diagnostic characters were present in the specimen examined, confirming the correct identification. Therefore, misidentifications of *P. maderensis* or a recent speciation event seem to be the most likely explanations, but

more sequences of specimens of both species would be needed to elucidate this point. In any case, the molecular cladogram does not even show sequence groupings according to the current taxonomy of the family Caristiidae.

Only a few species of *Lycodes* have been sequenced [48], making identification through DNA barcoding alone difficult. The sequence alignment of the Porcupine Bight specimen of *L. terraenovae* with those available in BOLD shows its closeness to *L. adolfi*, as previously reported [23]. However, the cladograms resulting from mitochondrial 12S and Cyt b markers show a close relationship with another specimen of *L. terraenovae*, in addition to *L. adolfi*.

Although there are obvious differences in the morphological characters between these two species, this relationship was already suggested in the first description of *L. adolfi* [49]. Molecular similarity could be due to misidentification, synonymy or recent diversification events. Allometric growth variability, sexual dimorphism and geographical variation have been found in some morphological characters of *L. terraenovae* [50,51] and Nielsen and Fosså [49] also found ontogenic changes in *L. adolfi*. This intraspecific morphological variability could lead to consider individuals of the same species as being of separate species and could be an important source of misidentifications and synonymies in this genus. An integrative and comparative study would be necessary with the aim to establish the true relationship between these two species.

Biodiversity requires vouchered and curated specimens for biomass measurements, morphological and genetic analysis (DNA barcoding) to confirm identification and establish robust taxonomy and phylogeny [18]. Molecular tools have been of increasing importance in the discovery of cryptic deep-sea species and taxonomic synonymies resulting from the phenotypic plasticity of a wide range of taxa. However, molecular analysis is a routine procedure, requiring comparatively less effort and formation than classical morphological taxonomy. Consequently, molecular data have grown considerably in the last decades in the repositories, whereas morphological analyses remain scarce in deep-sea fishes. Although modern molecular techniques allow plausible new taxonomic status to be proposed, the paucity of specialised taxonomic expertise and funding (taxonomic impediment [52]), means that these matters will remain unresolved for a long time.

Gaps and inconsistencies in reference DNA databases can make it difficult to accurately identify fish to the species level, suggesting the need to strengthen the DNA barcoding reference datasets [11]. There are considerable geographical variations in the number of deep-sea fish barcodes deposited in repositories. Higher activity has been detected in some countries, such as Canada and the United States in the Northwest Atlantic, and Australia and New Zealand in the Southeast Pacific. The Northeast Atlantic is comparatively less sampled, due to a lower tissue sampling effort in the European Atlantic countries, which is higher compared to less developed and resourced countries such as, for example, the African Atlantic coast.

This integrated approach is proving to be a powerful tool in the exploration of the diversity of the deep-sea, helping fill the gaps in the understanding of the true diversity of life in this environment.

**Supplementary Materials:** The following are available online at https://www.mdpi.com/article/10.3390/jmse9101075/s1, Figure S1. Neighbor-Joining trees based on p-distances, including COI-5P sequences of the 14 deep-sea species captured at Porcupine Bank. The percentage of replica trees in which the associated taxa clustered together in the bootstrap test (2000 replicates) are shown next to the branches when values are higher than 70%. The trees are drawn to scale, with branch lengths in the same units as those of the genetic distances used to infer the cladograms. Differences among sequences were computed as p-distances and are in the units of the number of base differences per site. All ambiguous positions were removed for each sequence pair (pairwise deletion option). There was a total of 652 positions in each final dataset. The analyses were conducted in MEGA X. Annotations are also included for each tree regarding the location where the specimens were collected. The species names shown in the cladograms are those that were associated with the sequences downloaded from BOLD. (A) *Paraliparis hystrix*; (B) *Platyberyx opalescens*; (C) *Lycodes*

*terraenovae*. Figure S2: Neighbor-Joining trees of *Lycodes terraenovae* mitochondrial Cyt b (A) and 12S rDNA (B) nucleotide sequences based on p-distances. The percentage of replica trees in which the associated taxa clustered together in the bootstrap test (2000 replicates) are shown next to the branches when values are higher than 70%. The trees are drawn to scale, with branch lengths in the same units as those of the genetic distances used to infer the cladograms. Differences among sequences were computed as p-distances and are in the units of the number of base differences per site. All ambiguous positions were removed for each sequence pair (pairwise deletion option). The analyses were conducted in MEGA X.

**Author Contributions:** Sampling: N.V.-A., C.F. and F.B.; conceptualization, R.B., A.d.C. and F.B.; methodology, R.B., A.d.C. and F.B.; formal analysis, R.B. and A.d.C.; writing—original draft preparation, R.B., A.d.C. and F.B.; writing—review and editing, R.B., A.d.C., C.F., N.V.-A. and F.B.; funding acquisition, F.B. All authors have read and agreed to the published version of the manuscript.

**Funding:** The Spanish Bottom Trawl Survey on the Porcupine Bank (SP-PORC-Q3) was funded in part by the EU through the European Maritime and Fisheries Fund (EMFF) within the Spanish National Program of collection, management and use of data in the fisheries sector and support for scientific advice regarding the Common Fisheries Policy.

**Institutional Review Board Statement:** Not applicable.

**Informed Consent Statement:** Not applicable.

**Data Availability Statement:** Not applicable.

**Acknowledgments:** The authors would like to thank the staff involved in the research survey PORCUPINE of the Spanish Institute of Oceanography (IEO) on board the R/V Vizconde de Eza (Ministry of Agriculture, Fisheries and Food, Spain).

**Conflicts of Interest:** The authors declare no conflict of interest.

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
