# Peer review of "Exploring Deep-Sea Biodiversity in the Porcupine Bank (NE Atlantic) through Fish Integrative Taxonomy"

_jmse, doi:10.3390/jmse9101075_

Round 1

Reviewer 1 Report

With a few exceptions, the authors considered my recommendations in the revised version. I believe the manuscript is now clearer and smother to read. However, I noticed that the authors made some significant changes to the results section and noticed that the text has not been entirely updated. Therefore, I recommend publication of this manuscript after some minor revisions (see comments below).

Line 19 – Replace “western” by “west”.

Line 19 - I would rephrase the beginning or the sentence: “A total of 14 fish species were identified, including: …”

Line 27 – I am a little confused here. I was under the impression that the results section had been modified and that it was 10 species that were consistent with morphological identifications, not 6.

Line 55 – Please replace “distribution” by “distributional”.

Line 56 – variations.

Line 82 – For more clarity, I think that it would be better to use the word “understudied” instead of “unreported”.

Line 83 – Please replace “far away to be complete” by “far from being complete”.

Line 177 – Please remove “species distribution” from the title of this section, since you are not studying species distribution.

Also, I maintain the comment I made on the previous version of this manuscript. I find this section difficult to read and still believe that a summary table would be a much better way to represent these data.

Line 317 – Section title missing (this should be section 3.2 I believe).

Line 398 – This sentence seems a bit awkward. Maybe replace “is search” by “was searched” or “was investigated”?

Line 528 – You are no longer representing 14 species, only three. Please correct here and in the supplementary file. Also, I think that a sentence should be added to explain why only three species are represented. Why is H. atlantica not represented since it is one of the taxa that was not consistent with morphological data?

Table S1 is not mentioned in the supplementary materials list.

Overall, please make sure to update the text based on your new results throughout the manuscript.

Figures – Figures resolution is really low but I am assuming that high-resolution figures will be included in the finale version of the manuscript?

Author Response

Line 19 – Replace “western” by “west”.

It is done

Line 19 - I would rephrase the beginning or the sentence: “A total of 14 fish species were identified, including: …”

This paragraph already do not appear in this version

Line 27 – I am a little confused here. I was under the impression that the results section had been modified and that it was 10 species that were consistent with morphological identifications, not 6.

The number of species has been corrected adequately

Line 55 – Please replace “distribution” by “distributional”.

We think “distribution” is also correct

Line 56 – variations.

This change has been done

Line 82 – For more clarity, I think that it would be better to use the word “understudied” instead of “unreported”.

In this case, the term "understudied " is not what we mean

Line 83 – Please replace “far away to be complete” by “far from being complete”.

 This change has been done

Line 177 – Please remove “species distribution” from the title of this section, since you are not studying species distribution.

This change has been done

Also, I maintain the comment I made on the previous version of this manuscript. I find this section difficult to read and still believe that a summary table would be a much better way to represent these data.

Line 317 – Section title missing (this should be section 3.2 I believe).

This is the section 3.2

Line 398 – This sentence seems a bit awkward. Maybe replace “is search” by “was searched” or “was investigated”?

This term already do not appear in this version

Line 528 – You are no longer representing 14 species, only three. Please correct here and in the supplementary file. Also, I think that a sentence should be added to explain why only three species are represented. Why is H. atlantica not represented since it is one of the taxa that was not consistent with morphological data?

H. atlantica is not represented because the name resulting on the BLAST search is a synonym, and therefore the same species

Table S1 is not mentioned in the supplementary materials list.

It is already mentioned

Overall, please make sure to update the text based on your new results throughout the manuscript.

The text has been revised

Figures – Figures resolution is really low but I am assuming that high-resolution figures will be included in the finale version of the manuscript?

Reviewer 2 Report

The manuscript ID "jmse-1395586" is very interesting and deserve to be considered for publication in JMSE. Integrative taxonomy reveals important news regarding the faunal assemblage of the Porcupine Bank. The paper is well written and organized. I have just few suggestions to improve the quality of the manuscript 

  • please provide a map for the study site
  • I would like to suggest to create tables to report the morphology of examined species, this could improve the readability of results. leave just the salient results that are considered in the discussion section.

All the best regards

the Reviewer 

Author Response

Most of the suggested changes in the text were made. Regarding the commentaries, R2 questioned the limited knowledge of deep-sea fishes, based on the increase in sampling and taxonomic revisions in recent decades. While it is true that a large number of taxonomic papers covering many families have been published in the last 30 years, this is clearly insufficient. We are familiar with all the revisions mentioned by this reviewer (Ceratioids, Pseudoscopelids, Stomiiformes, etc.) but many others remain to be revised (Moridae, Gaidropsaridae, Notacanthiformes, Liparidae, etc.). It is of particular interest to note that, in some cases, molecular data allowed us to intuit taxonomic aspects undetectable through examination of morphology. Although the sample size is small, this shows that the taxonomy of deep-sea fishes is far from complete and that research is needed to better understand the natural variability of the species through a greater sampling effort.

Reviewer 3 Report

This was an interesting paper and adds to understanding of the deep water fauna of the Porcupine Bank, as well as global distributions. The integrated approach of DNA and traditional morphology & meristics is not new, so I feel the authors are making a meal of it in this paper. I feel they could simply state that this is something they have done and the results confirm the value of such an approach. Certainly encouraging institutions to make collecting tissues as part of museological practice is to be lauded. However, there seems to be some issues around identification of some of the material, probably because the authors were not aware of some of the published revisions that have been done. Some of those have been included in the review. I would strongly recommend they look again where there is conflict between the DNA and their identification of the material. I have made some suggested changes to the writing style in the text to help the flow.

Author Response

A map for the study site was provided

This manuscript is a resubmission of an earlier submission. The following is a list of the peer review reports and author responses from that submission.

Round 1

Reviewer 1 Report

This study investigates deep-sea fish diversity in the Porcupine bank. In particular, the authors collected and identified fourteen fish species combining morphological examination and molecular DNA barcoding methods. In addition to providing valuable information in fish species frequenting the deep Porcupine bank, results from this study raise several inconsistencies in the phylogeny of several fish taxa, calling for additional taxonomy studies focusing on these taxa.

For these reasons I believe that this manuscript will be of interest to the scientific community and should be published after moderate revisions (see comments below).

Abstract

Line 18 – What do you mean by rare fishes? Are they rare worldwide? Or in the Porcupine bank? Please be more specific.

Line 19 – You could indicate that 14 species were identified.

Introduction

Lines 35 to 39 – For better flow, I suggest that you start the Introduction presenting the deep-sea in general (line 40) and mention the specific case of the Porcupine Bank later (line 78 for instance).

Line 38 – What do you mean by singular fish? How was their rarity defined? Based on low abundance? Limited distribution? Are they listed on the IUCN Red List? Please be more specific.

Line 52 – Please precise that they remain limited “in the deep sea”.

Lines 56 and 57 – No smooth transition between these paragraphs.

Line 57 – Remove “The” in front of “integrative taxonomy”.

Line 60 – References are missing.

Line 77 – This could be a good place to introduce Porcupine Bank and the need for a better knowledge of deep-sea fish diversity there.

Methods

Line 100 – How many trawl surveys did you do? What was the length and depth of each of these surveys?

I think that a map of the study area showing relevant trawl tracks would be useful here. If this is not possible, a table summarizing information on trawl surveys should be at least included.

Line 129 – I am curious, is there a reason you looked at additional markers for this species and not the others?

Line 149 – Please indicate how these 14 taxa were identified (through morphological examination).

Line 171 – I did not see any description of species distribution.

Lines 174 to 303 – I find this entire section extremely hard to follow. All the information is lost in the text and it takes a lot of effort to try get a good picture of the results presented. I think that it would be much clearer if all this information was presented in a table (one row per species).

A paragraph summarizing the main information regarding morphological characteristics and trends in species distribution should also be included.

Line 313 – I believe this figure (Figure S1) is very important and it is too bad to present it as a Supplementary Figure. It would be good to have a figure representing all the NJ trees within the article. For instance, a figure with 14 panels, one for each species. The photos of the specimens presented in Figure 1 could also be added to the relevant panels, next to the trees. That way you could remove Figure 1, leaving more space for a new, large Figure. Additionally, I suggest you use different colors in the trees when representing samples collected in the present study. That way it would be easier for the reader to visualize where your samples stand in the phylogeny.

Discussion

Line 357 – Please add relevant references.

Lines 450 to 456 – The reference to Figure S2 is missing.

Overall, I think that a discussion about fish diversity in the Porcupine Bank is required. Are the fish species you sampled representative of the Porcupine fish fauna? Are they rare? Widely distributed? Any information on their role in the ecosystem? Etc.

Figures and tables

General comment: why are figures and tables presented before they are mentioned in the text?

Figure 1 – Remove if you decide to include a figure representing results from the NJ analyses within the article. As previously mentioned, photos of the specimens could be included in that new figure to illustrate each tree.

Moreover, to shorten the legend, scale bars could be added next to each specimen.

Table 1 – This table could go into the Supplementary material. It does not need to be in the main article.

Table 2 and Figure 2 – I am not sure these table and figure are very useful and necessary here. To me it seems more important to keep space for a figure showing results from the NJ analyses.

Reviewer 2 Report

Review for Journal of Marine Science and Engineering

Exploring deep-sea biodiversity in the Porcupine Bank (NE Atlantic) through fish integrative taxonomy

JMSE-1241522

Precis: This manuscript details morphology, meristic and DNA barcode data for 14 deep-sea bony fish taxa from the Porcupine Bank in the north-eastern Atlantic.

Overall impression:  Although this MS contains some well-executed science, it is couched as something more than it really is – i.e. a research paper containing novel data and analyses whereas it really is a simple data paper. Here the authors have collected 14 species of fish, recorded standard morphological data and extracted and amplified the COI barcode gene. Subsequent to this they have then applied convoluted analyses which are not necessary. The discussion contains some speculative comments and some of the conclusions are not well-supported.

Comments: The methods are convoluted. Put simply it should have been: specimens caught, tissue sample taken, preliminary identification made based on overall morphology; morphological and meristic measurements taken and compared with literature for anomalous data; molecular data generated, barcode sequences compared with Genbank database (which harvests data from all barcode databases) using BLASTn for sequence match %.

At what point was a provisional fish species identification made? Surely the authors compared the specimens with the literature and decided at least on a putative identification? If so, how many were found to be correct using barcoding? It is mentioned in the discussion (line 354-356) however it then begs the question, why all the other analyses especially given the lack of conviction to the conclusions? Questions arising through analyses of molecular data are speculative and simply rely on more data to clarify – and that is all – there is no need for a large discussion on it.

The integrated taxonomic method is iterative and really intended for use at the point of species description by taxonomists. In other words, if there is no doubt regarding a species identification then the usual data for that particular group is provided in a description. If there is any doubt or there is conflicting information then data from an independent source is added. In modern taxonomy the standard is to use both morphological and molecular data so that both phenotype and genotyupe are covered. In the present paper both sources of data have been generated, however analysis has been applied no matter what. What the process of identification was is not clearly provided. In other words, in most cases there is no new data but merely more for some species.

There is always potential for GenBank data to be incorrect or mislabelled, or depositing authors may have used a provisional name for a species but have not amended it when a later analysis finds it to be misidentified. It is a mistake to assume that these are correct. Most of the phylogenetic reconstructions (trees) provided in this paper are unnecessary. A BLASTn result will suffice and then if the identification % is low then explore other identification analyses. Providing barcode gap % within the text without comparison is confusing, and not always necessary, especially where the identification is undoubted. A table is always better for presenting these data.

In some cases the authors found no barcode gap – this is because there will be no barcode gap with only one sequence. The ABGD process developed by Puillandre is the standard measure in taxonomy where doubts exist and these require 3 sequences per species. Drawing a conclusion for anything based on a singleton sequence is fraught with error. Essentially, where the pair-wise genetic divergence within a species is smaller than that between species then there is little or no doubt, if this is the case present it in a clear table form. In some of the species analysed here there simply isn’t enough data to do this analysis. Thus the barcode sequences ought only to be developed and provided for analysis for future studies – this has merit and is worthy of publication.

The meristic morphological data should be presented in table format for clarity. As it is in the text it is messy and complex. Table 2 is a much better way to present these data.

Table 1 – it really is only necessary to cite GenBank accessions as this harvests BOLD data. All data sourced from other sources should be presented in table format with the newly generated data so that readers do not need to read the figure captions to access these.

The interpretation of some cladograms is erroneous. The use of the expression ‘well-distant’ for some clades is not correct, given the scale of the trees. Clades may not be direct sisters but not being so does not equate to being ‘well-distant’. The polytomy in fig F has been misinterpreted. This tree certainly needs to be rooted to a moderately distant relative to obtain a grasp on distances. What are the bootstrap support values here? If they are low then throw out the tree. This brings us to the question of what data quality checks were carried out for barcode data? Were sequences checked for stop codon errors? How were alignments made, what substitution model was applied to generate trees?

Figure K – the tree for Caristiidae – the clustering of P. opalescens with P. maderensis does not necessarily indicate that P. madarensis has been misidentified. There is enough structure in these clades for them to remain separate albeit closely related sister taxa. There just isn’t enough data to make a call either way. It is no surprise that the tree does not reflect the deeper taxonomy of the Caristiidae – COI is a species delimiting marker. To explore genus and family level molecular structure other markers are needed in concatenation. It is entirely unnecessary to develop this tree that contains other genera in exploration of this species based on a singleton.

Lines 410 to 430 – there simply isn’t enough data here to synonymise P. hystrix and P. garmani. However the authors then write: ‘Although further analysis with more specimens would be necessary, the results of this research show enough evidence, both morphological and molecular, to propose P. hystrix as a junior synonym of P. garmani.” No. it needs to be one or the other.

Reviewer recommendation:

Reconsider after major revision.